# A Novel Antibody-Drug Conjugate Targeting Nectin-2 Suppresses Ovarian Cancer Progression in Mouse Xenograft Models

**DOI:** 10.3390/ijms232012358

**Published:** 2022-10-15

**Authors:** Yun Hee Sim, Yun Jung Um, Jeong-Yang Park, Min-Duk Seo, Sang Gyu Park

**Affiliations:** College of Pharmacy, Ajou University, 206 World Cup-ro, Yeongtong-gu, Suwon-si 16499, Gyeonggi-do, Korea

**Keywords:** nectin-2, ovarian cancer, chimeric antibody, antibody-drug conjugate

## Abstract

Ovarian cancer is the fifth leading cause of cancer, followed by front line is mostly platinum agents and PARP inhibitors, and very limited option in later lines. Therefore, there is a need for alternative therapeutic options. Nectin-2, which is overexpressed in ovarian cancer, is a known immune checkpoint that deregulates immune cell function. In this study, we generated a novel anti-nectin-2 antibody (chimeric 12G1, c12G1), and further characterized it using epitope mapping, enzyme-linked immunosorbent assay, surface plasmon resonance, fluorescence-activated cell sorting, and internalization assays. The c12G1 antibody specifically bound to the C2 domain of human nectin-2 with high affinity (K_D_ = 2.90 × 10^−10^ M), but not to mouse nectin-2. We then generated an antibody-drug conjugate comprising the c12G1 antibody conjugated to DM1 and investigated its cytotoxic effects against cancer cells in vitro and in vivo. c12G1-DM1 induced cell cycle arrest at the mitotic phase in nectin-2-positive ovarian cancer cells, but not in nectin-2-negative cancer cells. c12G1-DM1 induced ~100-fold cytotoxicity in ovarian cancer cells, with an IC_50_ in the range of 0.1 nM~7.4 nM, compared to normal IgG-DM1. In addition, c12G1-DM1 showed ~91% tumor growth inhibition in mouse xenograft models transplanted with OV-90 cells. These results suggest that c12G1-DM1 could be used as a potential therapeutic agent against nectin-2-positive ovarian cancers.

## 1. Introduction

Ovarian cancer is the second most common malignancy of the female genital tract, with an estimated burden of 210,000 global deaths in 2020 [1]. As more than 70% of patients are diagnosed at an advanced stage [2], the five-year survival rate of ovarian cancer patients after diagnosis is only 47% [3]. Ovarian cancer is classified into epithelial, sex-cord stromal, germ cell, and mixed types, according to the cell of origin [4]. Among these, about 90% of patients are diagnosed with epithelial ovarian cancer (EOC), and its histological subtypes include serous, endometrioid, clear cell, mucinous, malignant Brenner tumors, and mixed histologies, with over 60% of cases classified as serous histology [5,6]. High-grade serous carcinoma arises from a well-differentiated, low-grade serous carcinoma [7,8]. Recent advances in molecular characterization have shown that EOC can be further classified into two distinct groups: type I and type II [7,8,9,10,11]. The pathogenesis of type I EOC is mostly from endometriosis or fallopian tube-related serous borderline, which are characterized by genomic alterations, including those in KRAS, BRAF, PTEN, PI3K3CA, CTNNB1, and ARID1A. On the other hand, the pathogenesis of type II EOC mostly originates from precursor lesions in the fallopian tube epithelium that harbor TP53 mutation [4,12].

Treatment of ovarian cancer usually involves a combination of surgery and chemotherapy. The incorporation of targeted therapy, including PARP inhibitor and bevacizumab for EOC, has a significant impact on disease control [13,14,15]. Despite this progress, most women diagnosed with advanced EOC ultimately develop platinum-resistant recurrence [16]. Recently, targeted therapies, including those involving vascular endothelial growth factor and folate receptor alpha, have shown commendable progress in improving progression-free survival [17,18,19,20,21]. Nonetheless, there is still an unmet medical need to develop other therapeutics to efficiently treat patients with ovarian cancer.

Nectin-2 (CD112, poliovirus receptor-related protein 2) is a transmembrane glycoprotein that functions as a Ca^++^-independent cell adhesion molecule. It is composed of three immunoglobulin G (IgG)-like domains (one V domain and two C2 domains), a transmembrane domain, and a cytoplasmic domain [22]. The V domain of nectin-2 forms a cis- or trans-dimer to maintain cell-cell adherent junctions [23,24], which activates Cdc42 and Rac small G proteins, leading to the regulation of cellular events such as cell adhesion, movement, and polarization, via reorganization of the actin cytoskeleton [25]. It has been reported that nectin-2 is highly expressed in ovarian and breast cancers [22] and is significantly upregulated in ovarian cancer patients with lymph node metastasis [26]. Interestingly, vascular endothelial growth factor induces the downregulation of nectin-2 expression in peritoneal endothelial cells, resulting in increased endothelial permeability [26]. In addition, the nectin-2 overexpressed on cancer cells or antigen-presenting cells suppresses immune cell activation, as an immune checkpoint, by binding to the nectin-2 receptor (poliovirus receptor-related immunoglobulin domain-containing, PVRIG) that is expressed on the surface of cytotoxic T (CD8+ T cells) and natural killer (NK) cells. However, the underlying molecular mechanism remains under investigation. In addition, nectin-2 can bind to CD226 and TIGIT, which are expressed on T and NK cells [24,27]. While the binding of nectin-2 to CD226 activates immune cells, its binding to TIGIT suppresses the same [24,27].

The expression of mRNA and protein of nectin-2 is up-regulated in various cancer tissues, including breast, ovarian, and prostate cancer, compared to their adjacent non-tumor tissues [22,26]. It is over-expressed in the ovarian tumor biopsies irrelevant of histological subtypes, grading or distant metastasis [26]. It was shown to be nectin-2 positive in about 48% of ovarian cancer patients [22]. Surprisingly, higher gene expression of nectin-2 was found in tumors with lymph node metastasis compared to nodal negative tumors [26]. Recently, Oshima et al. showed that nectin-2 antibody with reduced antibody dependent cellular phagocytosis can be developed as therapeutic agents for ovarian cancer [22,28]. In this study, we generated and characterized a mouse monoclonal antibody targeting human nectin-2. In addition, we generated a chimeric antibody (c12G1) to develop the c12G1 antibody as an immune-oncology drug and immune check point inhibitor. However, the c12G1 antibody did not compete with PVRIG, the nectin-2 ligand, which indicates that c12G1 cannot be applied as an immune check point inhibitor. Thus, we generated a c12G1 antibody as a chimeric antibody-drug conjugate (ADC) and examined its therapeutic feasibility in ovarian cancer.

## 2. Results

### 2.1. Generation and Characterization of Anti-Nectin-2 Antibody

First, we examined nectin-2 expression in ovarian cancer cell lines. Quantitative real time—polymerase chain reaction (qRT-PCR)—and western blot showed that nectin-2 was overexpressed in ovarian cancer cells (Appendix A), in agreement with a previous report [22]. We then generated a mouse monoclonal antibody (m12G1 clone) and investigated its specific binding to nectin-2. Fluorescence-activated cell sorting (FACS) analysis showed that m12G1 antibody binds to various ovarian cancer cell lines, including OV-90, SK-OV-3, and Caov-3, in a dose-dependent manner, but not to Daudi cells, which is a nectin-2-negative cell line (Appendix A). In addition, a knock-down study using a nectin-2-specific small interfering RNA (si-RNA) resulted in reduced cell binding of the m12G1 antibody, which supports that the m12G1 antibody binds specifically to nectin-2 (Appendix A). The examination of quantitative binding affinity using surface plasmon resonance (SPR) assay showed that the m12G1 antibody can bind to human nectin-2 protein [equilibrium constant (K_D_) = 4.99 × 10^–10^ M] (Appendix A). We then generated a chimeric 12G1 antibody (c12G1) by grafting the variable domain of the m12G1 antibody onto human IgG1. FACS analysis showed that the binding of the c12G1 antibody to ovarian cancer cell lines was similar to that of m12G1 (Figure 1A and Appendix A). In addition, knock-down experiments further confirmed that the c12G1 antibody specifically binds to nectin-2 (Figure 1B). Furthermore, SPR analysis showed that there was no difference in the binding affinity to human nectin-2 protein (K_D_ = 2.90 × 10^−10^ M) (Figure 1C and Appendix A).

Recently, it has been shown that nectin-2 protein overexpressed in cancer cells functions as an immune checkpoint by binding to the nectin-2 protein receptor (PVRIG) of NK and cytotoxic T cells [29]. Cytokine secretion and cytotoxic activity of immune cells are suppressed upon binding of the V domain of nectin-2 with PVRIG. To address whether the c12G1 antibody can inhibit the interaction of nectin-2 and PVRIG and thereby be applied as an immune cell activator, we performed a competitive enzyme-linked immunosorbent assay (ELISA). As shown in Figure 2A, the c12G1 antibody did not interfere with the binding of nectin-2 to PVRIG, suggesting that the c12G1 antibody binds to sites other than the V domain of nectin-2. To identify the binding domain of the c12G1 antibody, we generated various deletion mutants of human nectin-2, as shown in Figure 2B. The immunoprecipitation assay after transfection of wild-type (WT), Δ1, or Δ2 mutants showed that the c12G1 antibody can bind to the WT and Δ1 mutant, but not to the Δ2 mutant, thereby indicating that the c12G1 antibody binds to the first C2 domain of nectin-2 (Figure 2C). In addition, since upon artificial overexpression, the nectin-2 protein could be structurally different from endogenous nectin-2 protein, we examined whether the c12G1 antibody can bind to endogenous nectin-2 protein. Immunoprecipitation assays using HEK293 cell lysates showed that the c12G1 antibody could bind to endogenous nectin-2 protein (Figure 2D).

In addition, the therapeutic efficacy of naked antibodies in oncology is determined in terms of their ability to bind antigens while simultaneously mediating effector functions, including complement-dependent cytotoxicity (CDC) and antibody-dependent cell-mediated cytotoxicity (ADCC). An in vitro effector function assay using the OV-90 cell line showed that the CDC and ADCC activities of the c12G1 antibody were not observed up to a concentration of 20 µg/mL (Appendix A). These results suggest that naked c12G1 antibody cannot be expected to have an anticancer effect by effector function, and thus, is not feasible for application as an immuno-oncology agent for the treatment of cancer.

### 2.2. c12G1 ADC Exhibits Anti-Tumor Activity In Vitro and In Vivo

Although nectin-2 is highly overexpressed in the OV-90 cell line, the c12G1 antibody cannot be used as an anti-tumor agent in the form of an antibody itself. Therefore, an ADC using the c12G1 antibody could be a reasonable option to effectively treat ovarian cancer. To address the feasibility of its application as an ADC, we investigated the internalization efficiency of the c12G1 antibody in ovarian cancer cells. FACS analysis showed that the internalization efficiency of the c12G1 antibody was 54.9% (OV-90), 59.1% (SK-OV-3), and 57.9% (Caov-3) (Figure 3A). It is known that the pH within lysosomes is slightly acidic [30]. Therefore, we further analyzed the trafficking of the c12G1 antibody/nectin-2 complex to lysosomes using anti-Fc FAB conjugated with Zenon™, a pH-sensitive fluorescent dye. As shown in Figure 3B, the internalization efficiency of c12G1 antibody was 81.4% in OV-90 cells, 45% in SK-OV-3 cells, and 28% in Caov-3 cells. Taken together, these results suggested that the c12G1 antibody could be used as an efficient carrier for the specific delivery of toxins to treat ovarian cancer. We then generated ADC using smcc-DM1, which comprised a non-cleavable linker and a microtubule inhibitor. Sodium dodecyl sulfate-polyacrylamide gel electrophoresis (SDS-PAGE) analysis showed that the conjugation of smcc-DM1 to the c12G1 antibody induced a slight size shift (Figure 4A). In addition, as DM1 and the antibody absorb ultraviolet light at 252 nm and 280 nm, respectively [31], we comparatively analyzed the absorbance of the naked antibody (c12G1) and ADC (c12G1-DM1) at 252 and 280 nm. The conjugation of DM1 to c12G1 increased the absorbance at 252 nm (Figure 4B). The drug-to-antibody ratio (DAR) was determined as described previously [32] and found to be approximately 5.07. Since the target-binding affinity of ADC can be reduced by a conformational change via conjugation of the N-hydroxysuccinimide ester of the smcc linker with the primary amines of lysine in the antibody, we compared the binding affinity of 12G1 antibody and 12G1 ADC. ELISA and FACS analyses showed that the binding affinities of the c12G1 antibody and c12G1 ADC to nectin-2 were similar (Figure 4C,D). These results indicated that conjugation of smcc-DM1 to the c12G1 antibody did not affect its binding affinity to nectin-2.

DM1 induces cell cycle arrest at the G2/M phase by inhibiting microtubule assembly and promoting apoptosis of actively dividing cells. Cell cycle analysis showed that c12G1-DM1 inhibited cell division at the G2/M phase, leading to an increase in sub-G1 population, apoptotic cells (Figure 5A). However, c12G1-DM1 did not affect the cell cycle in Daudi, the nectin-2-negative cell line. We then analyzed the cytotoxic activities of c12G1-DM1 in ovarian cancer cell lines. c12G1-DM1 exhibited in vitro cytotoxic activities in OV-90, SK-OV-3, and Caov-3 cells, with half-maximal inhibitory concentration (IC_50_) values in the hundred picomolar to nanomolar range (Appendix A). The in vitro cytotoxic activity of c12G1-DM1 against the nectin-2-positive cell lines was 5–300-fold higher than that against the nectin-2-negative cell lines (Figure 5B, Appendix A). Next, the in vivo efficacy of c12G1-DM1 was examined using mouse models xenotransplanted with the OV-90, SK-OV-3, and Caov-3 cells. In the OV-90 cell line, while the naked c12G1 antibody and IgG-DM1 did not suppress tumor growth, c12G1-DM1 suppressed tumor growth in a dose-dependent manner (Figure 5C). Additionally, 3 mg/kg and 5 mg/kg of c12G1-DM1 induced complete remission of the tumor for approximately 25 days (Figure 5C). In addition, while c12G1-DM1 did not show any anti-tumor activity in SK-OV-3, c12G1-DM1 inhibited tumor growth in a dose-dependent manner in Caov-3 cells (Figure 5C), except for in the case of one mouse in the 5 mg/kg treatment group that did not respond to administration from the initial phase of the treatment itself for unknown reasons (Appendix A). There was no change in the body weights of the ADC-treated mice (results not shown), and a single toxicity study of 20 mg/kg c12G1-DM1 using normal C57BL/6 mice did not affect the body weight change (Appendix A). Taken together, these results suggested that an ADC targeting nectin-2 could be useful in the treatment of ovarian cancer, as well as other nectin-2-positive cancers, including breast, colon, and lung cancers [22].

## 3. Discussion

It was reported that nectin-2 binds to PVRIG on NK and T cells, thereby limiting their anti-tumor activity [33]. To address the feasibility of its application as an immune checkpoint inhibitor, we investigated whether the c12G1 antibody could inhibit the interaction between nectin-2 and PVRIG and found that it did not (Figure 2A). While the V domain of nectin-2 is critical for interaction with PVRIG, the c12G1 antibody is associated with the first C2 domain of nectin-2 (Figure 2B,C), further confirming that the c12G1 antibody itself cannot be used as an immune checkpoint inhibitor. Therefore, we studied the feasibility of its application as an ADC in nectin-2-overexpressing ovarian cancer cells. In addition, our finding that the c12G1 antibody does not react with mouse nectin-2 protein (Appendix A) confirmed that it binds specifically to human nectin-2 protein, and therefore, cannot be applied to a mouse orthotopic model. A single toxicity study using normal C57BL/6 mice did not show changes in body weight at concentrations up to 20 mg/kg of c12G1-DM1 (Appendix A), thereby suggesting the possibility that c12G1-DM1 has no off-target binding.

In this study, we compared c12G1-smcc-DM1 and c12G1-vc-PAB-MMAE (Figure 4 and Figure 5, Appendix A). DM1 and MMAE, microtubule inhibitors, showed similar cytotoxicity between nanomolar and tens of nanomolar concentrations [34,35,36]. In our assay using ovarian cancer cell lines, smcc-DM1 and vc-MMAE also showed similar activity (results not shown). Although the DAR and target-binding affinity of c12G1-smcc-DM1 and c12G1-vc-PAB-MMAE were similar (5.07 vs. 4.79) and there is no bystander effect in smcc-DM1, smcc-DM1, which is a non-cleavable linker payload, exhibited much higher cytotoxic activity than vc-PAB-MMAE, a cathepsin B-cleavable linker payload (Figure 5B and Appendix A). Proteases, including cathepsin B, are highly expressed and secreted in EOC, resulting in metastasis via degradation of the extracellular matrix [37,38,39]. While smcc-DM1, a non-cleavable linker-payload, is released upon degradation of antibodies by various proteases in the lysosome, the cleavable linker can be cleaved in the extracellular environment by secreted cathepsin B, suggesting that the MMAE of c12G1-vc-PAB-MMAE can be released into the extracellular environment before internalization into cancer cells, thereby reducing its therapeutic efficacy. If the in vitro activity of a non-cleavable linker in ADC exhibits efficient therapeutic efficacy, this can be directly translated to in vivo efficacy because the degraded peptide of the antibody can boost immune cell activation near cancer cells. However, this requires further investigation. In addition, while amine coupling of the c12G1 antibody using smcc-DM1 exhibited a homogeneous band pattern in both non-reducing and reducing SDS-PAGE (Figure 4A), bridging of the free thiol group in the antibody with mc-vc-PAB-MMAE showed a heterogeneous band pattern in non-reducing SDS-PAGE, despite the homogenous band pattern observed in reducing SDS-PAGE (Appendix A). These results suggest that there are still free thiol groups in the antibody, after conjugation of mc-vc-PAB-MMAE, resulting in separation into heavy and light chains even in non-reducing SDS-PAGE and structural instability in solution.

Although SK-OV-3 shows a two-fold higher expression level of nectin-2 than Caov-3 (Appendix A and Figure 1A), a similar internalization percentage (Figure 3), and the cell growth rate of SK-OV-3 is higher than that of Caov-3 (Figure 5B), the in vitro cytotoxicity and in vivo antitumor activity of c12G1-DM1 are more sensitive in Caov-3 than SK-OV-3 (Figure 5B,C and Appendix A). One of the reasons for the above result can be attributed to multi-drug resistance. DM1 is a substrate of multidrug resistance protein 1 (MDR1, P-glycoprotein), which mediates the efflux of drugs from the cell [40]. However, it is known that the expression level of MDR1 is higher in Caov-3 than in SK-OV-3 [41], suggesting that the reduced anti-tumor activity of c12G1-DM1 in SK-OV-3 appears to be due to other causes that require further study.

Nectin-2 is overexpressed in various cancers, including breast, ovarian, melanoma, prostate, and pancreatic cancers [33,42]. Because nectin-2 overexpression in various cancers does not mean that nectin-2 is overexpressed in all the mentioned cancers [22], there is a requirement for the development of a companion diagnostic system, such as immunohistochemistry, to select nectin-2-positive cancers for application of targeted therapy. Patients with ovarian cancer are treated with platinum-based therapy, using a combination of paclitaxel, docetaxel, or cisplatin with carboplatin [43,44,45]. However, most patients ultimately develop platinum-resistant recurrence [16]. The selection of a nectin-2-positive cancer using companion diagnosis and determination of H-score, which quantitates the intensity and proportion of nectin-2 expression, would provide a better therapeutic option for ovarian cancer patients for the application of anti-nectin-2 ADC as a monotherapy or the combination of anti-nectin-2 ADC with chemotherapy. In conclusion, we developed a chimeric ADC targeting nectin-2, characterized its binding specificity, and examined its anti-tumor activity both in in vitro and in vivo models, which suggested that c12G1-DM1 is a potential therapeutic ADC that can be used to treat ovarian cancer.

## 4. Materials and Methods

### 4.1. Cell Lines and Culture

OV-90 cells were cultured in a 1:1 mixture of MCDB-105 medium (Sigma-Aldrich, St. Louis, MO, USA) and Medium 199 (HyClone, Logan, UT, USA) with 15% (*v*/*v*) fetal bovine serum (FBS, HyClone) and 1% (*v*/*v*) penicillin/streptomycin (P/S, HyClone). SK-OV-3 and Daudi cells were cultured in Roswell Park Memorial Institute-1640 medium (HyClone) with 10% (*v*/*v*) FBS and 1% (*v*/*v*) P/S. Caov-3, NIH-3T3, and HEK293 cells were cultured in Dulbecco’s modified Eagle medium with high glucose (DMEM with high glucose, HyClone), supplemented with 10% (*v*/*v*) FBS and 1% (*v*/*v*) P/S. All cells were incubated at 37 °C, in a humidified 5% CO_2_-containing incubator.

### 4.2. Antibody Generation

To generate a monoclonal antibody against nectin-2, human recombinant nectin-2 protein (50 μg, Sino Biological Inc., Beijing, China) was subcutaneously injected into 6-week-old female Balb/c mice using complete Freund’s adjuvant (Sigma-Aldrich). The subsequent immunization was repeated three times using the same amount of immunogen emulsified with incomplete Freund’s adjuvant (Sigma-Aldrich) at intervals of one week. A hybridoma was generated by means of fusion of immunized mouse splenocytes and SP2/0 cells, a mouse myeloma cell line, and screened in hypoxanthin, aminopterine, and thymidine (HAT) medium containing HAT supplement (Gibco, Grand Island, NY, USA) in DMEM medium. Isotyping was performed using the Beadlyte-Mouse Immunoglobulin Isotyping Kit (Upstate Co., Temecula, CA, USA). The amino acid sequence of the variable domain of the 12G1 clone was determined by means of nucleic acid sequencing, and a chimeric 12G1 (c12G1) antibody was generated by grafting the variable domain of mouse 12G1 antibody onto human IgG1. The c12G1 was subcloned into the pCHO 1.0 vector (Thermo Fisher Scientific, Waltham, MA, USA) and the recombinant plasmids were transiently transfected into CHO-S cells (Thermo Fisher Scientific) using OptiPRO™ SFM and FreeStyle™ MAX Reagent (both from Thermo Fisher Scientific), according to the manufacturer’s instructions. The antibody was purified using Protein A Sepharose and SP Sepharose columns (Invitrogen, Waltham, MA, USA), as described previously [32].

### 4.3. qRT-PCR

Total RNA was extracted from the cells using TRIzol™ reagent (Ambion^®^, Carlsbad, CA, USA), and cDNA was synthesized from it using ReverTraAce™ qPCR RT Master Mix with gDNA Remover (Toyobo, Osaka, Japan), according to the manufacturer’s instructions. For amplification of the target gene cDNA, 10 μM primer (Appendix A) and TOPreal™ qPCR 2X PreMIX with SYBR^®^ Green (Enzynomics, Daejeon, Korea) were mixed and reacted using the StepOne™ Real-Time PCR System (Applied Biosystems, Foster City, CA, USA). The expression level of the *NECTIN-2* gene in each cell line was normalized to that of *GAPDH*, using the 2^−ΔΔCt^ method.

### 4.4. Western Blot

Cells were washed twice with cold Dulbecco’s phosphate-buffered saline (DPBS; Lonza, Basel, Switzerland) and lysed with radioimmunoprecipitation assay buffer (150 mM NaCl, 50 mM Tris-HCl pH 8.0, 0.5% sodium deoxycholate, 1% NP-40, and 0.1% SDS) containing 1 mM phenylmethylsulfonyl fluoride (Sigma-Aldrich) and 1× protease inhibitor cocktail (Calbiochem, Darmstadt, Germany). The lysed cells were incubated on ice for 30 min and centrifuged at 21,000× *g* at 4 °C, for 20 min. Whole cell lysates were subjected to SDS-PAGE and transferred to polyvinylidene fluoride microporous membranes (Millipore, Burlington, MA, USA). Immunoreactive signals were detected using enhanced chemiluminescence solution (AbClon, Seoul, Korea). The following antibodies were used in this study: anti-nectin-2 (Abcam, Cambridge, UK), anti-FLAG (Medical & Biological Laboratory, Nagoya, Japan), anti-tubulin (laboratory-made), horseradish peroxidase (HRP)-conjugated anti-human IgG (Thermo Fisher Scientific), HRP-conjugated anti-rabbit IgG (Thermo Fisher Scientific), and HRP-conjugated anti-mouse IgG (Thermo Fisher Scientific).

For the immunoprecipitation assay, human NECTIN-2 gene constructs, including wild-type (WT; M1-V479), V domain deletion mutant (Δ1; M1-A31 + E167-V479), and V/C2 deletion mutant (Δ2; M1-A31 + S264–V479), were synthesized (Cosmo Genetech, Seoul, Korea) and subcloned into the pCMV6-FLAG vector (Origene, Rockville, MD, USA). Plasmids containing the NECTIN-2 construct were transfected into HEK293 cells using 150 mM NaCl and 10 µM polyethylenimine (Polysciences Inc., Warrington, PA, USA). HEK293 cells (2 × 10^6^ cells) were seeded into a 100-mm dish and cultured overnight. The cells were then harvested with NP-40 lysis buffer [150 mM NaCl, 0.5% NP-40, and 50 mM Tris-HCl pH 8.0] containing 1 mM phenylmethylsulfonyl fluoride and 1× protease inhibitor cocktail and lysed for immunoprecipitation. The whole cell lysate (1 mg) was incubated with c12G1 (10 µg) on a rotator at 4 °C, for 1 h. Following that, protein A/G beads (Santa Cruz Biotechnology, Dallas, TX, USA) equilibrated with NP-40 lysis buffer were added to the cells and incubated at 4 °C, for 1 h. The mixture was washed thrice with NP-40 lysis buffer, and the supernatant was completely removed after the last wash. Precipitated proteins were extracted and subjected to SDS-PAGE for western blot analysis.

### 4.5. si-RNA Study

To evaluate the specific binding of the c12G1 antibody to nectin-2, a knock-down experiment using si-RNA was performed. Nectin-2 si-RNAs were synthesized by Bioneer (Daejeon, Korea) (Appendix A). HEK293 cells (2 × 10^5^ cells/well) were seeded in a 6-well plate and cultured overnight. Next, 40 nM nectin-2 si-RNA mixture or control si-RNA (Santa Cruz Biotechnology) was transfected into HEK293 cells using Opti-MEM™ (Gibco) and Lipofectamine™ RNAiMAX (Invitrogen, Waltham, MA, USA) for 72 h, according to the manufacturer’s instructions. After 72 h, the HEK293 cells were harvested for flow cytometry, western blot, and qRT-PCR.

### 4.6. Flow Cytometry

Adherent cells (OV-90, SK-OV-3, Caov-3, NIH-3T3, and HEK293) were washed twice with DPBS and harvested using Cell Dissociation Buffer (Gibco) by incubating at 37 °C, for 5 min. The suspended cells (Daudi) were harvested and washed twice with DPBS. The cells (2 × 10^5^ cells per sample) were blocked with blocking buffer (cold DPBS containing 5% bovine serum albumin (BSA)) on ice for 1 h. The cells were then centrifuged at 1000× *g* and 4 °C for 3 min and stained with primary antibodies against c12G1, hIgG isotype (Thermo Fisher Scientific), c12G1-DM1, and hIgG-DM1, all diluted in wash buffer (cold DPBS containing 2% BSA). The cells were then washed twice with wash buffer and stained with the secondary antibody on ice for 1 h. Secondary antibodies, including Alexa Fluor™ 488-conjugated anti-mouse IgG (0.5 μg/mL; Thermo Fisher Scientific) and fluorescein isothiocyanate-conjugated anti-human IgG (0.3 μg/mL; Thermo Fisher Scientific), were diluted in wash buffer. Fluorescence signals were detected using CyFlow Cube 6 (Sysmex Partec, Gorlitz, Germany) and analyzed using the FCS Express™ 6 flow cytometry software (De Novo Software, Los Angeles, CA, USA).

### 4.7. SPR Assay

SPR analysis was performed using an SR7500DC (Reichert Technologies, Buffalo, NY, USA). Recombinant nectin-2 protein (10 µg, Sino Biological Inc.) was diluted with 20 mM sodium acetate buffer (pH 6.0) and immobilized on polyethylene glycol chips with 1× PBS pH 7.4, according to the manufacturer’s instructions. Various concentrations of c12G1 antibody diluted in PBS containing 0.05% Tween-20 were allowed to flow over the chips. The K_D_ was calculated using Scrubber2 software (Reichert Technologies).

### 4.8. ELISA

Human recombinant nectin-2 protein was diluted with 1× PBS (laboratory-made) to 20 ng/100 μL/well and coated onto a 96-well Immuno Clear Standard Module (Thermo Fisher Scientific) at 4 °C overnight. After blocking with 300 μL/well of blocking buffer (PBS containing 5% BSA) at RT for 2 h the plate was incubated with the following primary antibodies (100 μL/well): mouse anti-12G1, anti-c12G1, or anti-c12G1-DM1 diluted in PBS containing 0.1% Tween-20 and 1% BSA at RT for 1 h 30 min. The plate was washed four times with PBST and incubated with a secondary antibody (100 μL/well) at RT for 1 h. The secondary antibodies were HRP-conjugated anti-mouse IgG (1:1500 dilution, Thermo Fisher Scientific) and anti-human IgG (1:1500 dilution, Thermo Fisher Scientific) diluted in PBST. The plate was then washed four times with PBST and the sample was incubated with 100 μL/well of 1-Step™ Ultra TMB-ELISA Substrate Solution (Thermo Fisher Scientific) for 3 min. The reaction was terminated with 1 N H_2_SO_4_ (50 μL/well), following which the absorbance readings were taken at the wavelength of 450 nm, using a SPECTROstar^®^ Nano microplate reader (BMG Labtech, Ortenberg, Germany).

Competitive ELISA was performed to determine whether c12G1 could interfere with the binding of nectin-2 to PVRIG. Human recombinant PVRIG protein (100 ng/100 μL/well; Sino Biological Inc.) was coated onto the ELISA plate and each well was blocked as described above. Human recombinant nectin-2 protein (100 ng) was incubated with or without the 12G1 or hIgG isotype at RT for 1 h. After blocking, the plate was incubated with a pre-incubated mixture (100 μL/well) at RT for 1 h 30 min. The plate was washed four times with PBST and then incubated with 100 μL/well of rabbit polyclonal anti-nectin-2 antibody (1:500, Abcam) at RT for 1 h. HRP-conjugated anti-rabbit IgG (100 μL/well, 1:1500 dilution, Thermo Fisher Scientific) was added to each well at RT for 1 h. Then, the plate was washed four times with PBST and the HRP signal was measured as described above.

### 4.9. Effector Function Assay

For the ADCC assay, OV-90 cells, used as target cells (T), were harvested using Cell Dissociation Buffer and stained with 10 µM Hoechst 33,342 (Invitrogen) for 20 min. The stained cells were washed three times with DPBS and seeded onto a 96-well black cell culture microplate (Greiner Bio-One, Kremsmünster, Austria) at a density of 5 × 10^3^ cells per well. The cells were then treated with c12G1 at each of the indicated concentrations and incubated for 30 min. During incubation, human peripheral blood mononuclear cells (PBMCs; Cellular Technology Ltd., Cleveland, OH, USA), used as effector cells (E), were thawed in DMEM with low glucose (HyClone) containing 10% FBS and washed three times with DPBS. The viability of the PBMCs was over 90% when determined using a hemocytometer. The PBMCs were incubated with the target cells (E/T ratio, 30:1) in the presence or absence of c12G1. OV-90 cells were incubated at 37 °C for 6 h and counted using the Celigo Imaging Cytometer (Nexelom Bioscience, Lawrence, MA, USA).

For the CDC assay, OV-90 cells (5 × 10^3^/well) were seeded into a 96-well black cell culture microplate (Greiner Bio-One) and incubated overnight. c12G1 was then added to the plates and the cells were incubated with it for 30 min. Next, 20% (*v*/*v*) human serum complement (Quidel, San Diego, CA, USA) was added to the plate and incubated at 37 °C for 6 h. The cells were stained with 10 µM Hoechst 33342 for 20 min and counted using Celigo Imaging Cytometer.

### 4.10. Internalization Assay

Cells were blocked with Human BD Fc Block™ (BD Biosciences, San Diego, CA, USA) in DPBS containing 5% BSA at RT for 10 min. Then, c12G1 (1 μg/mL) diluted in cold serum-free media containing 2% BSA and 75 μg/mL cycloheximide was added to the cells, and the cells were incubated with it on ice for 1 h. After washing twice, the cells were resuspended in cold or pre-warmed serum-free medium containing 2% BSA and 75 μg/mL cycloheximide, and then seeded onto a 60-mm dish. Each sample was incubated at 4 °C or 37 °C for 1 h or 3 h. The cells were harvested and stained with fluorescein isothiocyanate-conjugated anti-human IgG (0.3 μg/mL; Thermo Fisher Scientific) diluted in wash buffer (cold DPBS containing 2% BSA) on ice for 1 h. After washing three times, the fluorescence signals were detected using CyFlow Cube 6 and analyzed using FCS Express™ 6 flow cytometry software. Additional internalization assay was performed using the pH-sensitive Zenon™ dye. OV-90 cells, SK-OV-3 cells, or Caov-3 cells (1 × 10^5^ cells) were seeded onto 48-well plates and incubated at 37 °C for 1 h. The cells were then treated with the c12G1 antibody/Zenon™ dye mixture (10:100 nM) at RT for 5 min and further incubated at 37 °C for 15 h. Following that, the cells were harvested using 0.25% Trypsin/1 mM EDTA (Welgene Inc., Gyeongsan, Korea) and centrifuged at 1000× *g* for 3 min at 4 °C. The cells were washed with cold DPBS and re-suspended in cold DPBS containing 0.1% BSA. Fluorescence signals were detected using CytoFLEX (Beckman Coulter Inc., Brea, CA, USA) and analyzed using CytExpert software (Beckman Coulter Inc.).

### 4.11. Generation of ADCs

Antibodies were dialyzed with BupH™ PBS Packs (Thermo Fisher Scientific) and conjugated with smcc-DM1 (MedChemExpress, Monmouth Junction, NJ, USA) at RT at a molar ratio of 1:20 for 1.5 h. The mixture was centrifuged at 21,000× *g* for 20 min at 20 °C, to remove aggregates. ADCs were purified using Sephadex G-25 PD-10 desalting column (GE Healthcare, Chicago, IL, USA) and stored in 10 mM sodium succinate, 6% (*v*/*v*) sucrose, and 0.05% (*v*/*v*) Tween-20 pH 5.0. The ADCs were further analyzed using SDS-PAGE and ultraviolet spectrometry. Analysis of the DAR ratio was performed as previously described [32].

### 4.12. Cell Cycle and In Vitro Cytotoxicity Analyses

Cells were seeded into a 96-well black cell culture microplate (for adherent cells, at a density of 3–8 × 10^3^ cells/well) or 6-well plate (for suspension cells, at a density of 1 × 10^5^ cells/well) and incubated in a humidified 5% CO_2_-containing chamber, overnight. The cells were then incubated with 1 µg/mL of the antibody or ADCs, for 24 or 48 h. The cells were fixed with 70% ice-cold ethanol at 4 °C, for 30 min, and washed twice with cold DPBS. Furthermore, the cells were stained with propidium iodide solution (50 µg/mL, with 0.1 mg/mL RNase and 0.05% Triton™ X-100) at 37 °C for 1 h. The cells were then documented using Celigo Imaging Cytometer and analyzed using FCS Express™ 6 flow cytometry software.

For the cytotoxicity assay, cells were seeded in a 96-well black cell culture microplate (2–5 × 10^3^ cells/well). The cells were then treated with antibody or ADCs at concentrations in the range of 0–400 μg/mL, and the plate was further incubated for 3–4 d, depending on the cell growth rate. When cell confluency reached 80–90%, the cells were stained with 10 µM Hoechst 33342 or 1 µg/mL Calcein AM (Invitrogen) for 20 min and counted using Celigo Imaging Cytometer. The IC_50_ values were calculated using Prism 5 (GraphPad Software Inc., San Diego, CA, USA).

### 4.13. In Vivo Experiments

The animal studies were approved by the Institutional Animal Care and Use Committee of Ajou University (IACUC approval number: 2021-0019). The experiments conformed to the Guidelines for the Care and Use of Laboratory Animals published by the United States National Institutes of Health. All experiments were performed in accordance with the relevant guidelines and regulations. To evaluate the toxicity of c12G1-DM1, vehicle or 20 mg/kg of c12G1-DM1 was intravenously administered to C57BL/6 mice (Koatech, Pyeongtaek, Korea), and the changes in body weight were monitored every 3 d for 15 d.

For xenograft assays, OV-90 (5 × 10^6^/100 μL), SK-OV-3 (5 × 10^6^/100 μL), and Caov-3 (5 × 10^6^/100 μL) cells were mixed with an equal volume of Matrigel^®^ (Corning, Corning, NY, USA) and subcutaneously injected into 5-week-old female C.B-17 severe combined immunodeficiency mice (Orient Bio Inc., Seongnam, Korea). When the average tumor volume reached 100–200 mm^3^, the mice were randomized into treatment groups for the efficacy study. Vehicle, c12G1, or ADCs diluted in DPBS were intravenously administered to the mice once a week, three times, at the indicated concentrations. The tumor dimensions were measured using a Vernier caliper, and tumor volumes were calculated from them, as follows:Tumor volume = (4/3) × π × (length/2) × (width/2) × (depth/2)The tumor growth inhibition rate was calculated as follows:Tumor growth inhibition rate =[1 − (RTV in the treated group)/(RTV in the control group) × 100 (%)]RTV = (tumor volume on the measured day)/(tumor volume on day 0)
where RTV is the relative tumor volume. At the end of the experiment, the mice were euthanized using CO_2_.

### 4.14. Statistical Analysis

Statistical analyses were performed using Prism 5. The data are presented as mean ± standard deviation. Statistical significance was calculated using an unpaired Student’s *t*-test with *Tukey’s post-hoc* test. The differences were considered statistically significant at *p* < 0.05.

## Figures and Tables

**Figure 1 ijms-23-12358-f001:**
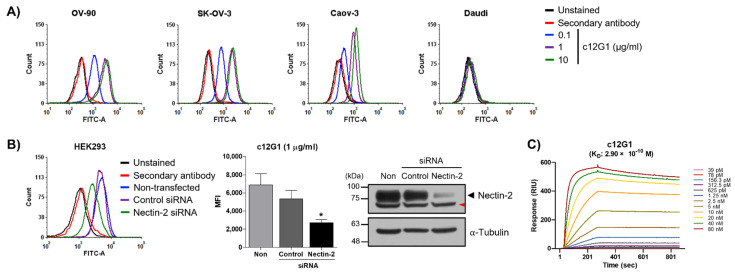
Characterization of the chimeric 12G1 (c12G1) antibody. (**A**) The binding of c12G1 antibody was determined using FACS analysis, at the indicated concentrations. Daudi cells were used as the nectin-2-negative cell line. (**B**) HEK293 cells were transfected with 40 nM of control or nectin-2 si-RNA, for 72 h. The FACS analysis was then conducted as described in the Methods section (*, vs. control si-RNA). Knock-down of nectin-2 expression was confirmed using western blot. Tubulin was used as the loading control. Anti-nectin-2 antibody from Abcam was used for western blot. The red arrowhead indicates a non-specific band. This experiment was independently repeated at least three times. (**C**) The binding affinity of c12G1 antibody to human nectin-2 was examined using SPR analysis. * *p* < 0.05.

**Figure 2 ijms-23-12358-f002:**
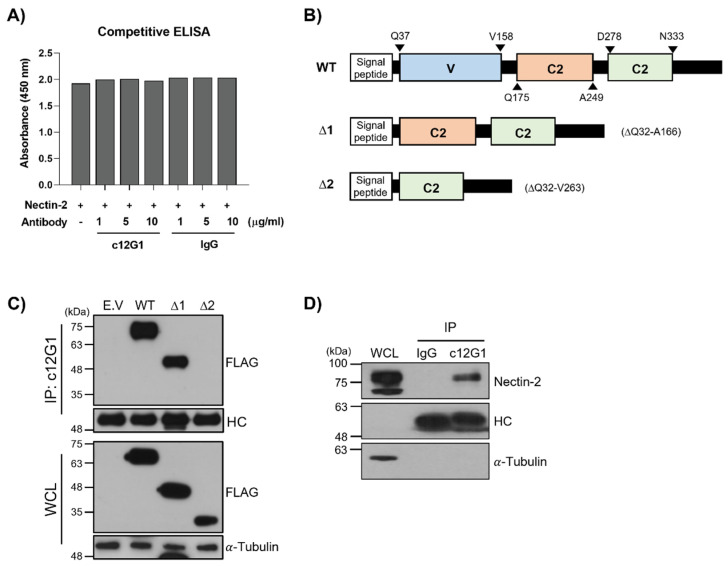
Identification of the nectin-2-binding domain of c12G1. (**A**) PVRIG (100 ng/well) was coated to 96-well plates and the binding of nectin-2 (1 µg/mL) to PVRIG was investigated in the presence of c12G1 antibody, at the indicated concentrations. The results represent mean ± SD of three independent experiments. (**B**) Schematic diagram for the construction of nectin-2 deletion mutants. The extracellular domain of the indicated nectin-2 deletion constructs was cloned into the pCMV6 FLAG vector. (**C**) The wild-type and deletion nectin-2 mutants were transfected into HEK293 cells. The whole cell lysates obtained from these cells were then subjected to immunoprecipitation assay using c12G1 antibody, followed by western blot with an anti-FLAG antibody. (**D**) To determine whether c12G1 antibody can bind to endogenous nectin-2 protein, the whole cell lysates of non-transfected HEK293 cells were subjected to immunoprecipitation assay using the c12G1 antibody or normal human IgG, followed by western blot with an anti-nectin-2 antibody. Ten percent (10 µg) of the cell lysates were loaded as the input control. Tubulin was used as the loading control. All experiments were independently repeated at least three times. HC indicates heavy chain of antibody.

**Figure 3 ijms-23-12358-f003:**
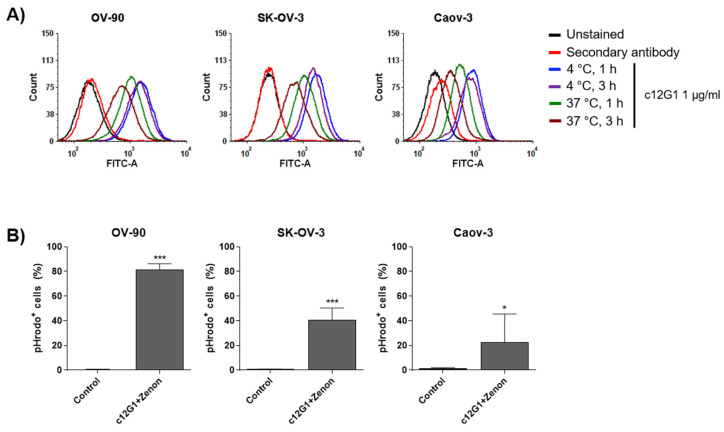
The c12G1 antibody is internalized into ovarian cancer cells. (**A**) Ovarian cancer cells were incubated in the presence or absence of c12G1 (1 µg/mL) at 4 °C or 37 °C, for 1–3 h, and then subjected to flow cytometer analysis. The fluorescence signal of the c12G1-nectin-2 complex on the cell surface decreased after incubation at 37 °C. (**B**) OV-90, SK-OV-3, or Caov-3 cells were treated with the c12G1/Zenon™-conjugated anti-Fc FAB complex for 15 h, following which the Zenon™-positive cells were analyzed using FACS (* and ***, vs. control). All experiments were independently repeated at least three times. * *p* < 0.05, *** *p* < 0.0001.

**Figure 4 ijms-23-12358-f004:**
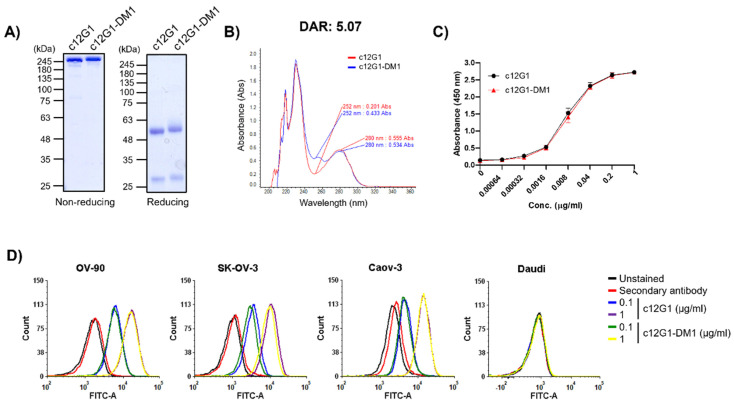
Characterization of c12G1-DM1. (**A**) The naked c12G1 antibody and c12G1-DM1 were compared using non-reducing and reducing SDS-PAGE. (**B**) The optical absorbance of c12G1-DM1 at the wavelength of 252 nm was compared to that of the naked c12G1 antibody. The calculated DAR was determined as 5.07. The binding affinity of naked c12G1 antibody and c12G1-DM1 to human nectin-2 protein was compared using ELISA (**C**) and flow cytometry (**D**). The nectin-2-binding affinities of naked c12G1 antibody and c12G1-DM1 were similar. Daudi cells were used as the nectin-2-negative cell line. All experiments were independently repeated at least three times.

**Figure 5 ijms-23-12358-f005:**
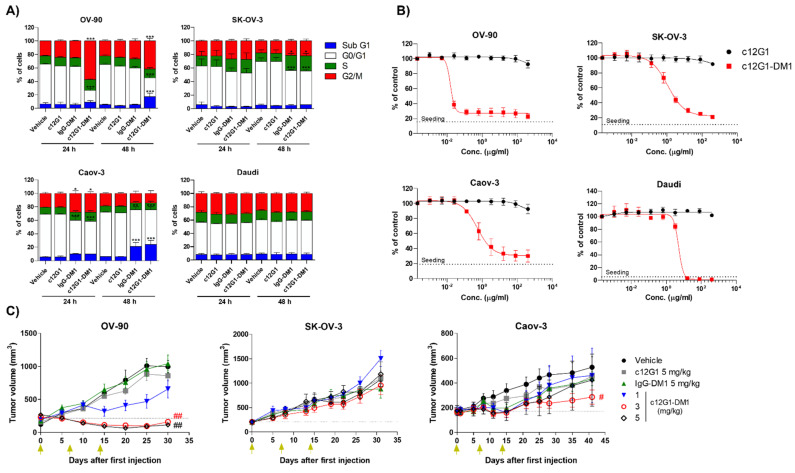
c12G1-DM1 exhibited anti-tumor activity, both in vitro and in vivo. (**A**) Ovarian cancer cells were treated with vehicle, c12G1 antibody (1 µg/mL), IgG-DM1 (1 µg/mL), or c12G1-DM1 (1 µg/mL), for 24 and 48 h. The cells were then fixed and stained with propidium iodide, followed by cell cycle analysis using a Celigo Imaging Cytometer (*, **, and ***, vs. their respective corresponding vehicle, c12G1, and IgG-DM1). c12G1-DM1 increased the cell population in the G2/M phase at 24 h, followed by an increase in the G1 population at 48 h. Daudi cells were used as the nectin-2-negative cell line. The results have been represented as mean ± SD from at least three independent experiments. (**B**) Cells were treated with serially diluted concentrations of the c12G1 antibody or c12G1-DM1, for 3–4 d. The cells were stained with 10 µM Hoechst 33342, at 37 °C for 30 min, and quantified using a Celigo Imaging Cytometer. The results represent mean ± SD of at least three independent experiments. The dashed line indicates the number of seeded cells. (**C**) Ovarian cancer cell lines were implanted into immunodeficient mice. The mice were randomized into different treatment groups when the tumor volume reached ~200 mm^3^ (n = 6), and then intravenously administered vehicle, c12G1, IgG-DM1, or c12G1-DM1, as indicated. Green arrows indicate the administration of vehicle, c12G1, IgG-DM1, or c12G1-DM1 (# and ##, vs. their respective vehicle, c12G1, and IgG-DM1). * *p* < 0.01, ** *p* < 0.001, *** *p* < 0.0001, ^#^
*p* < 0.01, and ^##^
*p* < 0.001.

## Data Availability

The data reported in this study are available from the corresponding author (sgpark@ajou.ac.kr) upon reasonable request.

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
