# Peer review of "A Novel Antibody-Drug Conjugate Targeting Nectin-2 Suppresses Ovarian Cancer Progression in Mouse Xenograft Models"

_ijms, 2022, doi:10.3390/ijms232012358_

Round 1

Reviewer 2 Report

Sim etal., manuscript entitled A Novel Antibody-Drug Conjugate Targeting Nectin-2 Suppresses Ovarian Cancer Progression in Mouse Xenograft Models.  In which Author generated and characterized a mouse monoclonal antibody targeting human nectin2. and also  generated a chimeric antibody to detect therapeutic aspects in ovarian cancer. Sim and co workers presented interesting work. The manuscript is well written and aptly described with necessary detailsHowever I have a point that must be addressed before paper will be suitable for publication.

explain the  uniqueness of your paper from below mentioned paper

Nectin-2 is a potential target for antibody therapy of breast and ovarian cancerMolecular Cancer 2013 Jun 12;12:60.doi: 10.1186/1476-4598-12-60.

Author Response

Reviewer 2

Q1. Explain the uniqueness of your paper from below mentioned paper.

Nectin-2 is a potential target for antibody therapy of breast and ovarian cancer. Molecular Cancer 2013 Jun 12;12:60.doi: 10.1186/1476-4598-12-60.

Answer: Basically, while Oshima et al tried to develop anti-cancer therapeutics using effector function, including ADCC as naked antibody, we initially tried to develop c12G1 antibody as an immune-oncology drug, immune check point inhibitor. However, c12G1 antibody did not compete with PVRIG, nectin-2 ligand, which indicates that c12G1 cannot be applied as an immune check point inhibitor. Thus, we intended to develop c12G1 as an ADC. To do this, we removed effector function, including ADCC and CDC, because synergistic cytotoxicity of payload with effector function can induce hypersensitivity reaction via hyper-activation of immune cells. In summary, there are basic difference from that of Oshima et al in terms of modality.

Round 2

Reviewer 2 Report

Author response was accepted and statements are need to be included in the manuscript. 

Manuscript will be suitable for publication.

Author Response

Q1. Author response was accepted and statements are needed to be included in the manuscript

Answer: We have corrected manuscript as you suggested. Statements are described in the last paragraph of introduction as followings ″ Recently, Oshima et al showed that nectin-2 antibody with reduced antibody dependent cellular phagocytosis can be developed as therapeutic agents for ovarian cancer [22,28]. In this study, we generated and characterized a mouse monoclonal antibody targeting human nectin-2. In addition, we generated a chimeric antibody (c12G1) to develop c12G1 antibody as an immune-oncology drug, immune check point inhibitor. However, c12G1 antibody did not compete with PVRIG, nectin-2 ligand, which indicates that c12G1 cannot be applied as an immune check point inhibitor. Thus, we generated c12G1 antibody as a chimeric antibody-drug conjugate (ADC) and examined its therapeutic feasibility in ovarian cancer.″